# Foraging Behavior Response of Small Mammals to Different Burn Severities

Marina Morandini [1,2,*], Maria Vittoria Mazzamuto [3] and John L. Koprowski [3]

1   Department of Natural Science, Paul Smith's College, Paul Smiths, New York, NY 12970, USA
2   School of Natural Resources and Environment, University of Arizona, Tucson, AZ 85721, USA
3   Haub School of Environment and Natural Resources, University of Wyoming, Laramie, WY 82071, USA; mariavittoria.mazzamuto@uwyo.edu (M.V.M.)
*   Correspondence: mmorandini@paulsmiths.edu

**Abstract:** Wildfires cause profound challenges for animals to overcome due to their reliance on vegetation. This study addresses the impact of three levels of forest burn severity (unburned, low, and high burn severity) on the foraging behavior of small mammals in the Pinaleño Mountains (AZ, USA) using the giving up density (GUD) experiment approach. Overall, burn severity affected the foraging behavior of small mammals that spent less time foraging in high burn severity patches. Vegetation characteristics influenced GUD differently based on the level of burn severity. Higher canopy cover was perceived as areas with a higher predation risk (higher GUD) in unburned and low burn severity patches, while cover provided by logs and shrubs decreased the GUD (increased foraging). This suggests a complicated interaction between horizontal (logs, grass, shrub cover) and vertical vegetation cover in relation to burn severity. Fires affected the foraging behavior of the small mammals but did not impact all species in the same way. Generalists, such as *Peromyscus* sp. and *Tamias dorsalis*, seemed to forage across all burn severities, while specialist species, such as tree squirrels, tended to avoid the high burn severity patches. Clarifying the complex impacts of fires on small mammals' foraging behaviors contributes to our understanding of the intricate interactions, at micro-habitat levels, between vegetation structure and the behavioral responses of animals and it can help managers to plan actions to reduce the negative impacts of wildfires.

**Keywords:** fire effects; behavior; GUD; predation risk; Mt. Graham

## 1. Introduction

Climate change is increasing the magnitude of extreme environmental events [1,2], such as wildfire severity and frequency [3–5]. In the western United States, fires are estimated to increase at a rate of seven additional large events each year [6]. This phenomenon is quite alarming considering the profound changes caused by fire in the composition and abundance of plant and animal species [7–9]. The direct effects of wildfires, such as death due to injury, extreme temperature, or smoke inhalation [10,11], are not an exhaustive description of the consequences of wildfires on wildlife. Indirect effects, including habitat loss and environmental changes, arise later, but can extensively impact animals' behavior and survival [10–12].

The largest obstacle animals have to overcome when their habitat is affected by fires is to suddenly deal with a new environment where the vegetation structure and composition have been modified and the quantity and distribution of resources have changed, while predation has potentially increased [11,13]. Changes in cover availability can influence the effective and perceived predation risk and subsequently alter the behavior, demographics, and growth rates of prey populations [14,15]. For example, periodically burned grassland fires influence herbivores' forage availability and quality, consequently affecting bison's (*Bison bison*) feeding behavior with an increase in bite mass and instantaneous intake rate per individual [16]. Woodpeckers (*Picoides* sp.) tend to select burned areas after a fire due

to an increase in insects in dead logs [17]. The ability to adapt to those changes is reflected in fitness, and ultimately in the entire population affected by fire [10,18].

The effects of fire on wildlife populations depend upon the ecology of the species [19–22]. For example, the population of the insectivore rodent *Akodon cursor* increased after the fire, whereas frugivore/granivore *Oecomys concolor* decreased or disappeared in some areas [23]. Usually, fires affect greatly isolated populations or populations with small geographic ranges, threatening their potential for recovery [24]. Fires also have a larger effect on habitat specialists, such as small to medium forest-dwelling mammals, because of their narrow niche breadth and reliance on specific vegetation community types [25].

Small mammals are a crucial component of forest ecosystems; they play an important role in the dispersal of plant seeds and spores of mycorrhizal fungi and are prey for many avian and mammalian predators [26,27]. Therefore, any changes in small mammal abundance and behavior due to disturbances, such as fires, may affect how the forest ecosystem functions. Fire creates a mosaic landscape of different degrees of burn severity, which influences the foraging behavior of rodents [28]. The post-fire landscape offers heterogeneity in opportunities and risks, with patches characterized by a higher concentration of resources, but also patches with high visibility (high predator encounter risk) due to the reduction in canopy and shrub cover. Predators do not only influence their prey by killing them, but also by creating a landscape of fear that influences the prey's activity time, foraging tactics, and microhabitat selection [29,30]. The experimental approach that has been long used to study spatial or temporal differences in animals' perception of benefit versus cost is the giving up density (GUD) model [31–34]. The GUD corresponds to resource density in a patch at which an animal stops to forage because further time spent in the same patch will add more costs (energetic costs, risk of predation, missed opportunity cost of not engaging in alternative activities) than benefits [33,35].

The increased landscape fragmentation caused by fire can radically change habitat connectivity for long periods of time [36], especially for small mammals as they perceive barriers to movement at finer spatial scales [37]. Several studies have investigated the influence of fire and fuel management treatments on small mammal abundance [13,26,38–40]. However, research focused on how disturbance affects animal behavior is scarce and the response of animals to burned patches is not always clear [41].

In this study, we addressed the impact of different burned patches on the foraging behavior of small mammals in the forested areas of the Pinaleño Mountains of southeastern Arizona, USA. In the summer of 2017, the Frye Fire was ignited on Mount Graham by a lightning strike and in 3 months burned 20,000 ha of forest, creating a mosaic landscape of different burn severity patches. In particular, (1) we analyzed the foraging decisions made by small mammals in different burned severity patches as a trade-off between costs and benefits; (2) we studied the role of vegetation characteristics in the cost–benefit analysis of foraging decisions; (3) we determined if different species show variation in the use of various areas with different levels of burn severity. The results provide insights into the multivariate nature of the foraging process and the diversity of factors upon which foraging decisions are made. Through an integrated approach using of giving up density experiments and camera traps, it was possible to understand how the post-fire landscape affects foraging behavior in small mammals, including the differential use of burned patches.

## 2. Materials and Methods

### 2.1. Study Site

The study area is located above 2750 m a.s.l. in the Pinaleño Mountains, Graham County, AZ, USA where the federally endangered Mt. Graham red squirrel (*Tamiasciurus fremonti grahamensis* [42]) occurs. The vegetation is dominated by Engelmann spruce (*Picea engelmannii*) and Rocky Mountain fir (*Abies lasiocarpa*) at the highest elevations, with Douglas fir (*Pseudotsuga menziesii)* and Southwestern white pine (*Pinus strobiformis*) occurring more frequently as elevation decreases [43].

In 2017, the Frye Fire burned 20,000 ha and ranged in elevation from 1219 to above 3000 m a.s.l., impacting the spruce-fir forest as well as the mixed-conifer forest. The Frye Fire created a mosaic landscape of different burn severity patches [44]. Within this mosaic, we studied the effect of burn severity on small mammals from May to July during the two years following the fire (2018 and 2019), at elevations of 2750 m and above. We specifically selected this elevation range to include the habitat of the endangered Mt. Graham red squirrel. We selected 4 areas (Grant Hill, Soldier Trail, Bible Camp, and the vicinity of the Mt. Graham International Observatory) that were affected by the Frye Fire in the summer of 2017 and that were easily accessible.

### 2.2. Study Design

With the ArcGIS version 10.8.1 software we created a polygon of about 2 ha around each of the selected 4 study areas. Within each polygon, we generated 50 random points at least 50 m apart. We overlapped these points with the aerial image of the area post-fire (fire perimeter and burn severity were obtained from the US Forest Service Burned Area Emergency Response Program) and classified them into three categories of fire severity. We then randomly selected 5 points for each burn severity category within the area. Therefore, each area included 15 patches, 5 for each distinct burn severity. A patch was an area with a 10 m radius attributed to a burn severity category, following the classification by Parson et al. (2011) [45]. Class 1 "unburned" was categorized by almost all surface organics (soil, woody debris, surface roots, etc.) remaining intact with little to no charring, with vegetation remaining green, and tree canopies being unaltered. Classes 2 and 3 were identified under the term "low/moderate severity" and consisted of surface organics consumed leaving brown to black charring and gray ash, vegetation going from green to brown from scorching, and canopy foliage scorched but not completely consumed. Class 4 "high severity" was the most severe with almost all surface organics destroyed with heavy charring and more ash, with almost no vegetation remaining, and canopy foliage completely consumed. The distance between these patches was much smaller than the home range of a single small mammal and it allowed individuals to visit multiple patches during their daily movements [46–48].

### 2.3. Vegetation Analysis

To investigate microhabitat features that could affect the foraging behavior of small mammals in addition to burn severity, we established a 10 m radius plot around the tray at each patch location, and we collected vegetation data both in 2018 and 2019 (Table A2). In each plot, we measured the diameter at breast height (DBH; cm) of all woody plants (shrub DBH < 10 cm, tree DBH > 10 cm; [49]), we recorded the species, and we classified the tree as alive and unburned, dead, or damaged by the fire. At 5 m and 10 m intervals from the center of the plot, we measured canopy density in each of the 4 cardinal directions, using a spherical densiometer and following the method proposed by Strickler (1959) [50]. For each piece of coarse woody debris (CWD) (defined as wood longer than 1 m with a diameter >20 cm), we recorded its length, smallest diameter, and largest diameter. The volume of CWD was determined using Smalian's volume formula. To estimate the grass and shrub cover in each plot, we first recorded the percentage of each in the 4 quadrants and, subsequently, we averaged the values. Shrub cover was considered as any vegetation at least 40 cm tall (including ferns and small trees).

### 2.4. Giving up Density Experiment

We determined how the different burn severities affected the foraging behavior of small mammals using the giving-up density (GUD) experiment [31,32] in the 15 patches of each study area. We used one plastic tray (58 × 41 × 16 cm) placed in the center of the patch (2 L capacity each, 5 trays in each patch type, and 15 trays each study area) filled with 2 L of play sand mixed with 50 g of black oil sunflower seeds. We used sunflower seeds

because have largely been utilized in similar experiments with many different rodents (*Peromyscus* sp. [51–54]; *Tamiasciurus* sp. [55,56]).

Trays also contained a lattice-like mesh to make the search for seeds more difficult. The seed trays were set out in the field for 3 consecutive days between May and July 2018 and 2019. In 2019 two repetitions were completed in each area. We considered the first day as a pre-baiting session, to let animals become familiar with the artificial food and tray, whereas we used the second and third days to measure the GUD. We sifted the trays once a day before sunset, and the sunflower seeds that remained in the tray were collected and replaced with another 50 g of seeds. Each time the sifted seeds were weighed with a precise digital milligram scale with a measurement range of less than 50 g and a precision sensor system of 0.001 g with an error within 0.005 g.

### 2.5. Burn Severity Patches Used by Different Species

We used a camera trap (Bushnell Trophy camera—model 119436) in video mode to enable the identification of the species visiting the trays [57]. From each video, we recorded the species, the day, and the time. Cameras were placed 40 cm above the ground facing the tray and set up to record 15 s videos, with 45 s intervals between consecutive videos. Cameras were active 24 h per day, and the batteries were checked every day.

### 2.6. Statistical Analysis

We performed all statistical analyses with R (Version 4.0.3; R Development Core Team, Auckland, New Zealand). Before proceeding with the data analysis, we averaged the grams of seeds retrieved in the tray (GUD) of the two consecutive nights of the experiment to avoid pseudo-replication. We applied a two-step analysis: we first assessed whether small mammals were more or less likely to visit the tray based on the burn severity of the patch where the tray occurred and, on the year, when the experiment occurred. Then, just for those trays that were visited, we assessed how the GUD was affected by the same two factors (burn severity and year). We investigated the presence/absence of any small mammals at the trays using a generalized linear model with binomial distribution and logit link function. We considered no seeds eaten as the absence of small mammals (value = 0), whereas when seeds were eaten from the tray (GUD < 50 g) we considered it as a presence (value = 1). The explanatory variables tested were year (2018 and 2019) and patch type (unburned, low burn severity, high burn severity), whereas the area was treated as a random factor. Next, we subset the data, considering only the trays visited by small mammals. We fitted a general linear model with Gaussian distribution and identity link, where the explanatory variable and random variables were the same as in the previous model. Model assumptions were verified by plotting residuals versus fitted values.

To determine which microhabitat characteristics influenced GUD in the 3 different burn severity patches, we fitted a linear model, grouping data by the burn severity of the patch. We verified possible correlation among the vegetation characteristics using the Pearson correlation test ($r \geq |0.7|$). Area was included as a random factor, while the total volume of logs, percentage of grass cover, shrub cover, and canopy cover were included as fixed effects after being scaled.

We also assessed if the number of species present at each tray differed among patches with different burn severity using a generalized linear mixed model (family Poisson, log link), with area as a random effect and patch type as a fixed effect. For this analysis, we did not include large mammals detected by the camera (bear, fox, skunk) or birds (for a complete description of species detected by the camera see Table A1). We also qualitatively analyzed the absolute frequency of each species counting the number of trays they used in each patch type across the 2018 and 2019 field seasons.

## 3. Results

The presence of animals in the trays was lower in 2018 (65% of the trays visited) than in 2019 (all but one tray visited); however, the presence of small mammals was not affected

by burn severity (Table 1). For the trays that were visited, GUD did not differ between years but was affected by the burn severity of the patch (Table 2). In fact, more grams of seeds were left in completely burned patches than in partially or unburned patches (Figure 1).

**Table 1.** Results of the generalized linear mixed model (binomial distribution, logit link), examining the effect of burn severity and year on the GUD (grams of seeds left at the tray) in the Pinaleño Mountains, Graham County, AZ, USA, in 2018 and 2019.

| Variable | B | SE | z | p |
|---|---|---|---|---|
| Intercept | 0.58 | 0.65 | 0.89 | 0.37 |
| Year (2019) | 437.80 | 105.20 | 4.16 | **<0.0001** |
| Unburned | −0.00002 | 0.67 | 0 | 1 |
| Low burn severity | 0.47 | 0.69 | 0.69 | 0.49 |

Statistically significant results in bold.

**Table 2.** Results of linear mixed model (Gaussian distribution, identity link), examining the effect of burn severity, year, and total volume of logs on the GUD (grams of seeds left at the tray) in the Pinaleño Mountains, Graham County, AZ, USA, in 2018 and 2019.

| Variable | β | SE | t | p |
|---|---|---|---|---|
| Intercept | 31.04 | 4.52 | 6.86 | 0.0009 |
| Year 2019 | −2.29 | 2.19 | −1.04 | 0.29 |
| Unburned | −7.13 | 2.31 | −3.08 | **0.002** |
| Low burn severity | −5.63 | 2.29 | −2.46 | **0.01** |

Statistically significant results in bold.

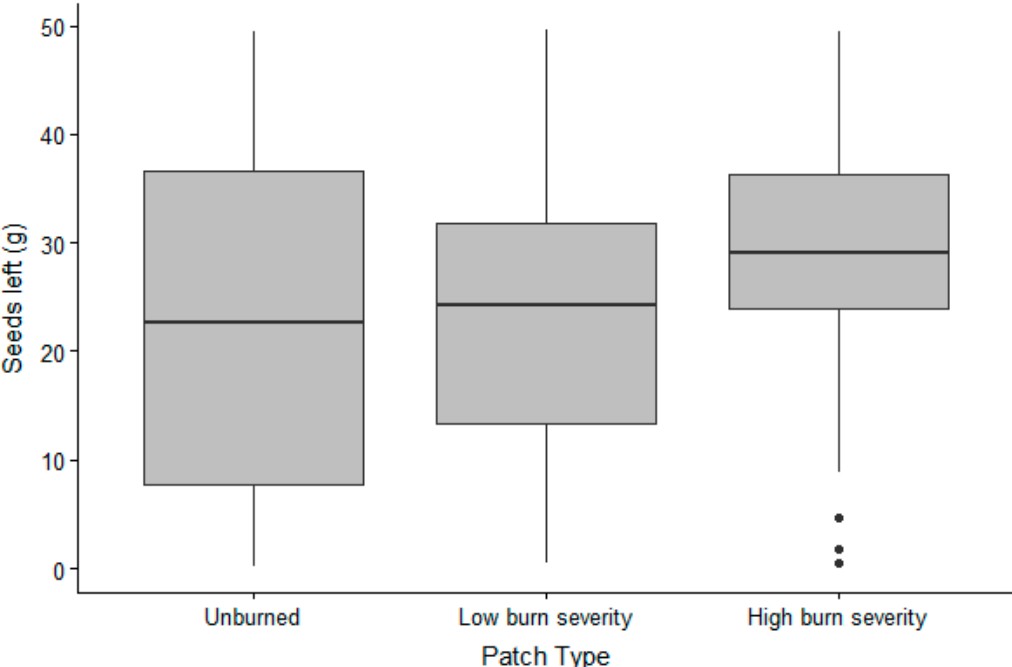

**Figure 1.** Grams of sunflower seeds collected from the trays (therefore non-eaten by animals = high GUD) in the 3 burn severity patches in the Pinaleño Mountains, Graham County, AZ, USA, in 2018 and 2019.

Vegetation characteristics affected GUD differently based on the patch type (Table 3). GUD was higher in unburned patches with little shrub cover but high canopy cover (Figure 2a–c). In the low burn severity patches, GUD was higher with high canopy and grass cover (Figure 2d). In the high burn severity patches, GUD was lower when there were more logs on the ground (Figure 2e).

**Table 3.** Results of linear mixed models, examining the effect of vegetation characteristics on the GUD (grams of seeds left at the tray) in different burn severity patches in the Pinaleño Mountains, Graham County, AZ, USA, in 2018 and 2019.

| Variable | Unburned | | | | Low Burn Severity | | | | High Burn Severity | | | |
|---|---|---|---|---|---|---|---|---|---|---|---|---|
| | β | SD | t | *p* | β | SD | t | *p* | β | SD | t | *p* |
| Intercept | 15.19 | 4.67 | 3.24 | 0.008 | 22.48 | 3.21 | 6.99 | 0.005 | 29.27 | 2.98 | 9.82 | <0.001 |
| Volume logs (m$^3$) | −8.62 | 3.07 | −2.80 | **0.007** | 2.49 | 1.31 | 1.90 | 0.06 | −8.76 | 3.08 | −2.84 | **0.006** |
| % Grass cover | 0.06 | 1.58 | 0.04 | 0.97 | 2.04 | 2.11 | 0.96 | 0.33 | 0.82 | 1.49 | 0.55 | 0.58 |
| % Shrub cover | −4.50 | 1.35 | −3.33 | **0.002** | 1.32 | 2.76 | 0.48 | 0.63 | −0.21 | 4.59 | −0.05 | 0.96 |
| % Canopy cover | 9.17 | 4.09 | 2.23 | **0.03** | 8.75 | 2.67 | 3.26 | **0.002** | 1.34 | 2.33 | 0.58 | 0.56 |

Statistically significant results in bold.

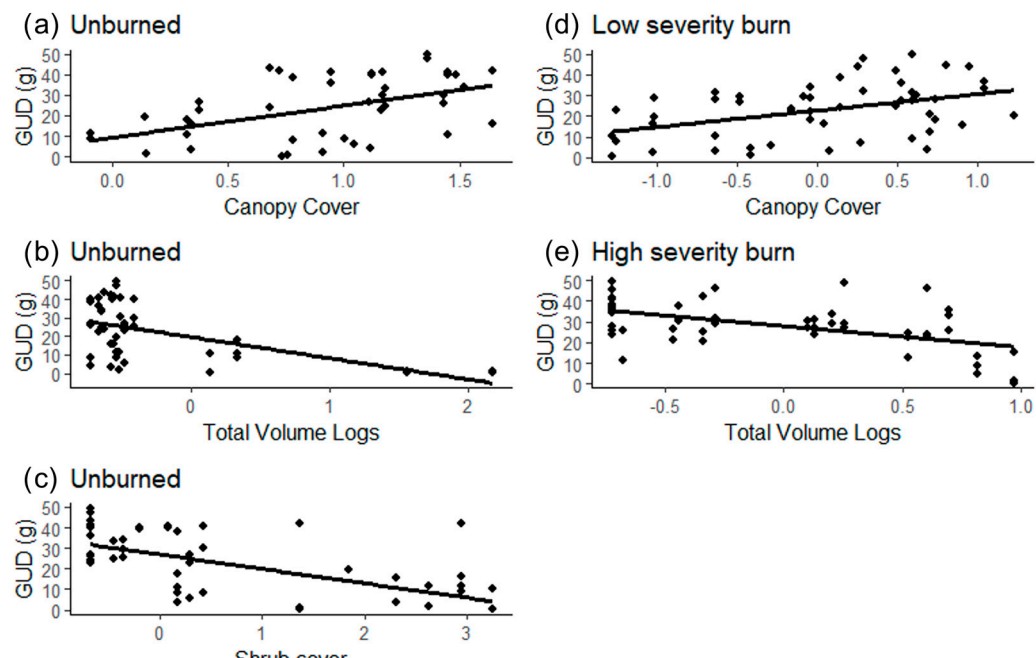

**Figure 2.** Effect of statistically significant vegetation characteristics on the GUD (grams of seeds left at the tray) in different burn severity patches [(**a**–**c**) unburned, (**d**) low severity burn, (**e**) high severity burn] in the Pinaleño Mountains, Graham County, AZ, USA, in 2018 and 2019.

We detected the following species visiting the trays: Abert's squirrel (*Sciurus aberti*), cliff chipmunk (*Tamias dorsalis*), Mt. Graham red squirrel (*Tamiasciurus fremonti grahamensis*), mouse (*Peromyscus* sp.), Mexican woodrat (*Neotoma mexicana*), rock squirrel (*Otospermophilus variegatus*), long-tailed vole (*Microtus longicaudus*), gray fox *(Urocyon cinereoargenteus)*, black bear (*Ursus americanus*), bird (species unknown), and striped skunk (*Mephitis mephitis*) (Table A1).

The different burn severity patches were used by a different number of species (Figure 3), with a mean of 1.72 (SD 1.03) species for unburned areas, 1.36 (SD 0.59) for partially burned areas, and 1.22 (SD 0.48) for completely burned areas. Species number was higher in unburned patches (beta = 0.55, 95% CI [0.17, 0.94], *p* = 0.005), compared to totally burned patches while there was no difference between the number of species detected in low burn severity patches versus totally burned patched (beta 0.23, 95% CI [−0.18, 0.64], *p* = 0.267), and between unburned and low burn severity (beta 0.32, 95% CI [0.05, 0.70], *p* = 0.09).

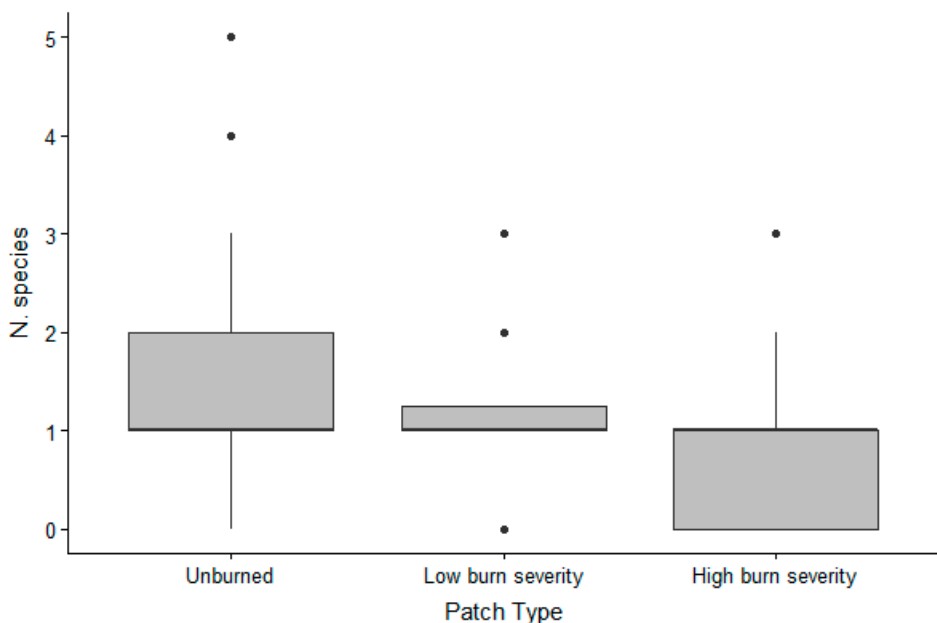

**Figure 3.** Box plot of the number of species of vertebrates detected in each tray per burn severity in the Pinaleño Mountains, Graham County, AZ, USA, in 2018 and 2019.

We observed a different species composition in the 3 categories of burned areas, although dominated by chipmunks and mice in all 3 patch types. Woodrat and Mt. Graham red squirrel used primarily unburned areas, whereas voles and Abert's squirrels were only present in unburned areas (Figure 4).

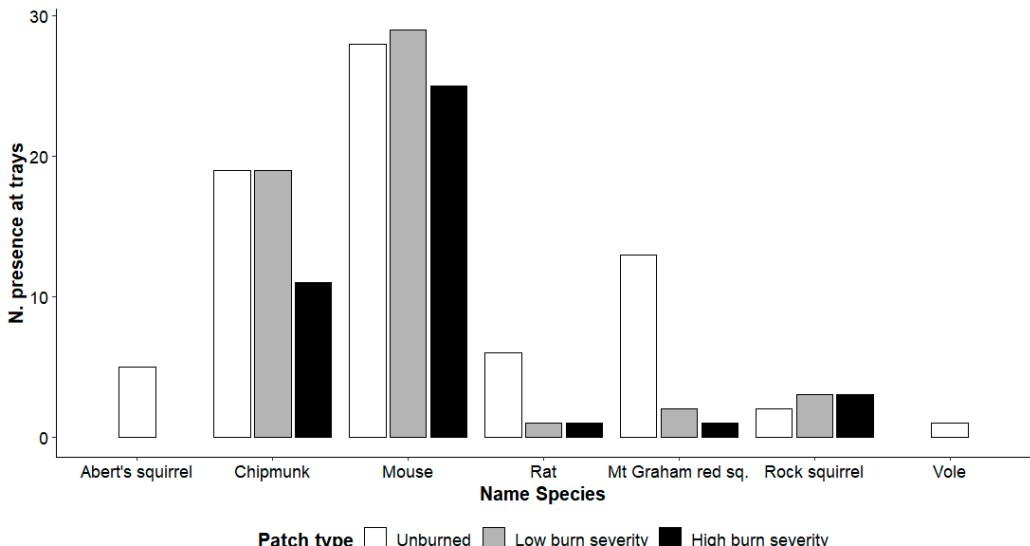

**Figure 4.** Number of absolute frequencies of presence at the trays for each species in each patch type (unburned, low burn severity, high burn severity) in the Pinaleño Mountains, Graham County, AZ, USA, in 2018 and 2019. The total number of trays is 40 per patch type (20 each year).

## 4. Discussion

Varying levels of burn severity influence small mammal foraging behavior. In high burn severity patches, small mammals stopped foraging in trays earlier than in low burn severity and unburned patches. The larger amount of seeds left indicated greater costs than benefits in remaining to eat for longer times. The most known cost associated with foraging is predation risk. Vegetation characteristics in patches of different burn severities

can influence the perception of predation risk [58] and consequently decisions regarding time spent foraging in a single location. In this study, we found that an increase in canopy cover corresponds to a lower number of seeds eaten, while an increase in horizontal cover (volume of logs and shrub cover) corresponds to a larger quantity of seeds eaten. We also found that fire affected the collective foraging behavior of small mammal communities but did not impact all species the same way, causing a smaller impact on generalist species.

The composition and abundance of small mammals varies as a function of time since the fire event [59] and depend upon the successional stages as well as the possibility of recolonization from nearby unburned habitats [60]. Immediately after a fire, populations can decline due to mortality [11,61], but the total small mammal biomass can increase in the two years after the fire event [26]. The higher presence of small mammals at the tray in 2019 suggests that the rodent populations in the study areas rebounded 2 years after the fire. In contrast with the presence/absence of mammals in patches, GUD is instead affected by differences in the level of burn severity, but not by year. Foraging behavior results in a complex response where benefits and costs are evaluated to maximize the energy intake. The main cost associated with high intensity burn areas is the risk of predation, generally perceived to be higher in the absence of vegetation cover [28,52,62]. Previous studies showed that rodent activity and occurrence are related to indirect (microhabitat) and not direct (olfactory and visual) cues of predation risk [63–65]. Thus, the presence of small mammals is more related to factors such as ground cover, refuges, and visibility to reduce the risk of predation than to the actual presence/activity of predators. As a result, GUD in high burn severity patches increased (foraging decreased) compared to unburned patches.

Predators create a landscape of fear that influences activity times, foraging tactics, and microhabitat selection of prey [29,66]. Foraging behavior may be influenced by different elements when considering microhabitats in different burn severity patches. In unburned areas and low burn severity patches, tree canopy cover has an important influence on GUD. However, we found that as canopy cover increased, GUD increased. Horizontal cover provides concealment from avian predators [67], but vegetation can also reduce the visual detectability of predators by prey [68,69]. Another possible explanation of this result is the presence in the Pinaleño Mountain of birds of prey that are adapted to hunt below the canopy of mature trees, such as Cooper's hawk (*Accipiter cooperii*), sharp-shinned hawk (*A. striatus*), Mexican spotted owl (*Strix occidentalis lucida*), or great-horned owl (*Bubo virginianus*). With the canopy cover above the predators, prey species are no longer directly protected. Instead, logs or cover offered by lower vegetation can provide immediate refuge from predation [28] by birds of prey, as well as terrestrial predators such gray foxes (*Urocyon cinereoargenteus*) and bobcats (*Lynx rufus*) present in our study site. In fact, in unburned areas, an increase in shrub cover and volume of logs corresponded to a decrease in GUD, hence higher removal rates of seeds from the trays. A similar pattern was found in high burn severity patches where log volume significantly affected GUD, with more seeds removed from the trays when a larger volume of logs was present. The positive relationship between canopy cover and GUD, but the negative relationship between cover offered by horizontal cover (logs, shrub cover) and GUD, aligns with the explanation provided by Potash et al. [58,70], who demonstrated that prey could perceive different predation risks as a consequence of interactions between multiple environmental cues in heterogeneous landscapes [58,70]. This heterogeneity is created by a different distribution of predators in the landscape, creating spatial variation in a prey's fear [71], and the interaction between vegetation cover along horizontal and vertical axes. In the absence of data collected in the same areas before the fire event, we cannot assess whether the small mammal response to vegetation changed after the fire.

The GUD experiment provides us with information on the assemblage of small mammals, but it does not provide information on the variability in gathering and/or feeding behavior by species. Thanks to the pairing of camera traps and trays we assessed that, while fire did affect the foraging behavior of small mammals, the effects varied among

species. A lower number of species used the trays in high burn severity patches than in unburned patches. Fire did influence small mammal populations; however, the level of effect was not uniform and appeared to be associated with specific habitat requirements of individual species [72,73], and the home range size of the animal. In this context, we expected generalist species to be less impacted, whereas habitat specialists negatively affected [74,75]. We observed the use of all types of burn severity patches by the generalist *T. dorsalis* and *Peromyscus* spp., whereas voles and Abert's squirrels were present only in unburned areas, and Mt. Graham red squirrel used mostly non burned areas, rarely visiting trays in other patch types. Previous studies showed the behavioral response of Mt. Graham red squirrel to fire, with an increased home range size and maximum distance traveled [25,46]. After the Frye Fire, surviving animals were forced to travel within and among isolated patches of live trees for cone harvesting and in search of new territories, leaving them more susceptible to predation [25]. In other studies, mice did not always modify their behavior as a consequence of disturbances or habitat types. For example, the white-footed mouse (*Peromyscus leucopus*) did not change its patterns of habitat use in response to fuel reduction treatments [76], and the deer mouse (*Peromyscus maniculatus*) did not change the quantity of foraging activity between habitats characterized by different shrub density [77].

## 5. Conclusions

Fire influences behavioral response in communities of small mammals, with high burn severity patches used by fewer species and perceived as riskier than unburned or lower burn severity patches. Vegetation variables play a different role in the foraging decision of small mammals showing a complicated interaction between horizontal (logs, grass, shrub cover) and vertical vegetation cover in relation to burn severity [58]. As wildfires threaten animals globally [78–80], and future increases in fire frequency and severity in the southwest USA will increase the loss of forest areas and potentially exacerbate the impact of predators on small mammals [81], understanding the relationship between animal foraging behavior, burn severity, and microhabitat can help managers plan actions to reduce the negative impacts of wildfires [28]. Fuel treatment is an important management technique in forests to reduce habitat loss in the long term; however, in the short term, it can exacerbate the impact on small mammals because of a lack of logs and shrubs that help mitigate the landscape of perceived fear, a factor that should be considered by forest managers when species of conservation concern are involved (e.g., the Mt. Graham red squirrel). Moreover, with the small mammals' role in the food web, changes in small mammal populations can also affect predatory species of conservation concerns (e.g., *Strix occidentalis* [82], *Martes caurina* [83]). Therefore, while fuel treatment is a tool that can help forest-associated species over the long term by reducing wildfires and the consequent habitat loss and fragmentation, forest managers need to balance the long-term benefit with the risk of short-term impacts on small mammals and their predators [84].

**Author Contributions:** Conceptualization, methodology, data collection, data analysis and original draft preparation—M.M.; review and editing—M.V.M.; review, supervision, funding acquisition—J.L.K. All authors have read and agreed to the published version of the manuscript.

**Funding:** This research was funded by Arizona Game and Fish Department, (grants no. I18005 and I16002), and T & E Inc. Grants for Conservation Biology.

**Institutional Review Board Statement:** All field work was conducted under the University of Arizona Institutional Animal Care and Use Committee protocol # 16-169, the Arizona Game and Fish Department scientific collecting permit # SP651773 for 2019, SP403044 for 2020, SP407072 for 2021, the U.S. Fish and Wildlife Service permit # TE041875-2, and adhered to the American Society of Mammologist's guidelines for the use of wild mammals in research (Sikes and Gannon, 2011).

**Informed Consent Statement:** Not applicable.

**Data Availability Statement:** The data presented in this study are openly available in FigShare at 10.6084/m9.figshare.20740765.

**Acknowledgments:** We would like to thank the Mt. Graham Red Squirrel Research Program graduate and undergraduate research assistants for valuable help in the field. This manuscript was improved by comments from R. W. Mannan, L. Wauters, and R. Steidl. Thanks to five anonymous reviewers who improved the initial version of the manuscript.

**Conflicts of Interest:** The authors declare no conflict of interest.

## Appendix A

**Table A1.** Number of trays where each species was detected during the experiment (60 trays for three rounds) and percentage of each species detected over the total number of trays during the entire experiment (180 trays) on the Pinaleño Mountains, Graham County, AZ, USA, in 2018 and 2019.

| Species | N Detection | Percentage |
|---|---|---|
| *Peromyscus* sp.—Mouse | 82 | 45.55% |
| *Tamias dorsalis*—Cliff chipmunk | 49 | 27.22% |
| *Tamiasciurus fremonti grahamensis*—Mt. Graham red squirrel | 16 | 8.88% |
| *Mephitis mephitis*—Striped skunk | 10 | 5.55% |
| *Otospermophilus variegatus*—Rock squirrel | 8 | 4.44% |
| *Neotoma mexicana*—Mexican woodrat | 8 | 4.44% |
| *Ursus americanus*—Black bear | 6 | 3.33% |
| *Sciurus aberti*—Abert's squirrel | 5 | 2.77% |
| Birds | 4 | 2.22% |
| *Urocyon cinereoargenteus*—Gray fox | 1 | 0.55% |
| *Microtus longicaudus leucophaeus*—Long-tailed vole | 1 | 0.55% |
| No species detected | 30 | 16.66% |
| NA (problems with camera, but seeds eaten) | 25 | 13.88% |

**Table A2.** Mean and standard deviation (SD) for each vegetation characteristic in the three different burn severity patch type in the Pinalegno Mountain in 2018 and 2019.

| Patch Type | Year | % Canopy Cover | | % Shrub Cover | | % Grass Cover | | Volume of Logs m$^3$ | |
|---|---|---|---|---|---|---|---|---|---|
| | | Mean | SD | Mean | SD | Mean | SD | Mean | SD |
| Unburned | 2018 | 83.72 | 10.81 | 8.92 | 9.15 | 19.99 | 22.32 | 133,374.45 | 217,664.65 |
| | 2019 | 78.97 | 10.96 | 11.33 | 10.98 | 20.05 | 22.31 | 133,374.45 | 216,208.68 |
| Low burn severity | 2018 | 69.63 | 11.03 | 0.86 | 1.72 | 12.56 | 13.01 | 208,708.30 | 225,619.18 |
| | 2019 | 54.95 | 17.44 | 4.15 | 5.44 | 20.45 | 16.84 | 306,591.92 | 420,116.44 |
| High burn severity | 2018 | 38.43 | 21.47 | 0.71 | 1.50 | 17.36 | 20.29 | 169,837.83 | 164,172.06 |
| | 2019 | 38.43 | 21.33 | 2.25 | 2.73 | 17.23 | 17.77 | 169,837.83 | 163,129.69 |

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
