# Peer review of "Foraging Behavior Response of Small Mammals to Different Burn Severities"

_fire, doi:10.3390/fire6090367_

Round 1
Reviewer 1 Report (Previous Reviewer 1)
Nice job responding to previous review comments.
Author Response
Thank you!
Reviewer 2 Report (New Reviewer)
This work is aimed to explore the impacts of variuos levels of forest burn severity (unburned, low, and high burn severity) on the foraging behavior of small mammals in the Pinaleño Mountains (Arizona, USA) using the giving up density (GUD) experiment approach. The paper has a good idea; however authors should improve their manuscript. I have major revision fo this paper.
Line 127 and 128, should be deleted.
All tables should improve for their quality.
All italic names must be revised “in italic form”
The following sentence should be blew the table 1 and the same in table 2,3.. “Area was entered as a random effect in the model. The model's R2 = 0.61
Figure 2 should be organized and defined each figure (A, B, C…etc).
In figure 4, why birds species are not presented?
Tables should put blew their results.
Line339, (Peromyscus leucopus) should be italic and the same in line 341.
Conclusion has huge numbers of citations
I recommended briefing the conclusions. Try to point massages for readers.
Please, References should be revised well.
References aren’t following the journal gridlines.
minor corrections have been found.
Author Response
This work is aimed to explore the impacts of variuos levels of forest burn severity (unburned, low, and high burn severity) on the foraging behavior of small mammals in the Pinaleño Mountains (Arizona, USA) using the giving up density (GUD) experiment approach. The paper has a good idea; however authors should improve their manuscript. I have major revision fo this paper.
Line 127 and 128, should be deleted.
All tables should improve for their quality.
All italic names must be revised “in italic form”
The following sentence should be blew the table 1 and the same in table 2,3.. “Area was entered as a random effect in the model. The model's R2 = 0.61
Figure 2 should be organized and defined each figure (A, B, C…etc).
Tables should put blew their results.
Line339, (Peromyscus leucopus) should be italic and the same in line 341.
ANSWER: all revisions above accepted
In figure 4, why birds species are not presented?
ANSWER: birds were not the target of the study and detections are by-catch data that are not representative of their populations. Camera traps with ground seed feeders were not set up to survey birds. Other types of surveys should be used to target bird populations.
Conclusion has huge numbers of citations. I recommended briefing the conclusions. Try to point massages for readers.
ANSWER: the conclusions were shortened and focused.
Please, References should be revised well. References aren’t following the journal gridlines.
ANSWER: corrected
This manuscript is a resubmission of an earlier submission. The following is a list of the peer review reports and author responses from that submission.
Round 1
Reviewer 1 Report
This paper describes a fairly straightforward analysis of the relative abundance of small mammals foraging from a baited station in three classes of burned plots (non-burned [NB], partially burned [PB], and completely burned [CB]. The results and discussion appear superficially reasonable and consistent with expectation, yet I found myself wondering about the soundness of results because of missing details. Specifically, the analysis relies entirely on relative frequency of detection of individual species; what about absolute abundance? Could you see similar trends if absolute abundance of most species remains constant while abundance of mice and chipmunks alone varies considerably with burn intensity? Can you conclude anything about the rarer species from relative abundance? Also, what assumptions are being made regarding relationships among species present and seed consumption? Are all species assumed to be equivalent as consumers? We know that species exhibit considerable variability in gathering and/or feeding behavior, so to what extent is this considered in the analysis of GUD? Finally, the discussion seems to suggest that managers are likely to be concerned about providing hiding cover to small mammals when implementing fuel treatments. Seriously? I can imagine more reasonable scenarios where forest managers are far more interested in reducing wildfire hazard and possibly improving habitat conditions for predatory species such as spotted owls than worrying about abundance of mice and chipmunks.
Lastly, the paper needs better proofreading. I noticed many grammatical errors.
Reviewer 2 Report
My primary reservation with the results is that no data and apparently no work was done on potential predators that would contribute to the perceived predation risk. The cameras picked up grey fox, but no other observations on potential avian predators is presented. If the authors have anything that can fill this gap the ms would benefit.
The authors discuss fire and post-wildfire conditions but I would encourage them to extend their perspectives on the role of habitat structure and how vegetation, downed wood, etc. provide cover to the wider debate around salvage logging of wildfire areas and clear cut logging and the need to retain structure. This could easily be accomplished with 3-4 sentences.
The manuscript needs a thorough grammar and spelling edit. For example, l. 30. Reads “In western United State, fires are…” should be “In the western United States, fires…”
There are many such errors in the ms that I cannot take the time to edit. Also, on l. 49, The genus sp. is usually italicised and in parentheses. This may be the format for the journal but I wasn’t sure as this is the first ms I have looked at for the journal Fire. Please have the authors do a thorough spelling and grammar check before re-submitting. Errors like Non Burned that should be Non-Burned (in Table 3, col 1) are relatively trivial but annoying.
Reviewer 3 Report
This manuscript investigates the effects of different fire severities on the foraging behavior of small mammals, by using the giving up density approach. The authors hypothesize that high severity fires will increase the perceived predation risk for small mammals, although ecological attributes of each species may lead to different responses to fire severity. The theoretical background behind the idea of the paper is quite solid, as several studies have already demonstrated that fire has indirect effects on small mammals, mediated through changes in habitat structure and the corresponding amplified predation risks (see Leahy et al. 2016). And indeed, the idea of testing the effects of fire on species foraging response is quite interesting, as most studies have addressed the effects of fire on community parameters. However, I think the manuscript has a few major issues that need to be addressed:
1) A major problem, in my opinion, is the definition and classification of each habitat patch in the three burn-severities patches discussed throughout the paper. The evaluation of fire severity seems a little arbitrary and subjective, weakening the results obtained and precluding future replications of the experiment elsewhere . This is more problematic for the partially burned category. In my opinion, it would be more valid if you choose and index to evaluate fire severity. See Roberts et al. (2008 -Roberts, S.L., van Wagtendonk, J.W., Miles, A.K. et al. Modeling the Effects of Fire Severity and Spatial Complexity on Small Mammals in Yosemite National Park, California. fire ecol 4, 83–104 (2008). https://doi.org/10.4996/fireecology.0402083) who used satellite imagery compiled by Thode (2005) and others to map fires and then used the NBR (normalized burn ratio) to define burn classes. See also Culhane et al. Small mammal responses to fire severity mediated by vegetation characteristics and species traits. Ecol Evol. 2022 May 19;12(5):e8918. doi: 10.1002/ece3.8918. PMID: 35600681; PMCID: PMC9120878. for other ideas.
2) The introduction could be more concise, and you could focus on how fire could change the behavior of small mammals (I know there are few studies). But I think you should focus on the importance of fire indirect effects and the increased perceived predation risk (there is a list of references in this issue).
3)Your second hypothesis needs to be clarified. What do you mean by the prediction that arboreal species are more affected by differential burned severity of the vegetation?
4) There are repeated sentences in the last paragraph of the introduction and in the second paragraph of the methods (79-82 and 103-107). Revise
5) The study design (2.2) was not clear to me, especially the part regarding the identification of the patches and the selection of experimental plots for the GUD experiment. Can you clarify?
6) Foraging experiment: Did you checked in a pilot study if all the species consumed black oil sunflower seeds? Do you any data on the abundance of the species in each of the patches before the fire?
7) Vegetation analysis - Any data on the vegetation of the patches before the fire?
8) Results: If, by any chance, the vegetation structure was different between the patches even before fire, your results could be flawed. Hence, it is possible that species richness and abundance differ between patches, independent of fire effects? The same question is valid for the analysis of vegetation attributes. Moreover, if this could be true, the observed differences in foraging responses could be due to different abundance of each species in each patch, and not due to increased perceived predation risk due to higher fire severity. I think you need to think how to deal with this caveat (failing to present data from before fire) in your study
9) In my opinion, the authors need to deal with the study caveats during the discussion. Moreover, I am not sure if it is worth insisting on the idea of testing the effects of fire severity. Maybe it is safer to test foraging responses of small mammals in burned x unburned sites. It is a simple question, but by choosing this alternative, authors could escape some problems of the study presented so far.